# Neonatal Plasma Exosomes Contribute to Endothelial Cell-Mediated Angiogenesis and Cardiac Repair after Acute Myocardial Infarction

**DOI:** 10.3390/ijms24043196

**Published:** 2023-02-06

**Authors:** Xiuya Li, Yilin Lian, Yukang Wu, Zihui Ye, Jiabao Feng, Yuan Zhao, Xudong Guo, Jiuhong Kang

**Affiliations:** 1Clinical and Translational Research Center of Shanghai First Maternity and Infant Hospital, Shanghai Key Laboratory of Maternal Fetal Medicine, School of Life Sciences and Technology, Tongji University, Shanghai 200092, China; 2Shanghai Key Laboratory of Signaling and Disease Research, Frontier Science Center of Stem Cell Research, National Stem Cell Translational Resource Center, School of Life Sciences and Technology, Tongji University, Shanghai 200092, China; 3Institute for Advanced Study, Tongji University, Shanghai 200092, China

**Keywords:** myocardial infarction, neonatal plasma exosomes, endothelial cells, angiogenesis, exosome proteome

## Abstract

Acute myocardial infarction (AMI) accompanied by cardiac remodeling still lacks effective treatment to date. Accumulated evidences suggest that exosomes from various sources play a cardioprotective and regenerative role in heart repair, but their effects and mechanisms remain intricate. Here, we found that intramyocardial delivery of plasma exosomes from neonatal mice (npEXO) could help to repair the adult heart in structure and function after AMI. In-depth proteome and single-cell transcriptome analyses suggested that npEXO ligands were majorly received by cardiac endothelial cells (ECs), and npEXO-mediated angiogenesis might serve as a pivotal reason to ameliorate the infarcted adult heart. We then innovatively constructed systematical communication networks among exosomal ligands and cardiac ECs and the final 48 ligand–receptor pairs contained 28 npEXO ligands (including the angiogenic factors, Clu and Hspg2), which mainly mediated the pro-angiogenic effect of npEXO by recognizing five cardiac EC receptors (Kdr, Scarb1, Cd36, etc.). Together, the proposed ligand–receptor network in our study might provide inspiration for rebuilding the vascular network and cardiac regeneration post-MI.

## 1. Introduction

Ischemic heart disease, such as myocardial infarction (MI), has long been a leading cause of death worldwide. MI causes scarring due to the cell cycle cessation of adult cardiomyocytes (CMs) and lack of resident cardiac stem cells [1]. Despite advances in revascularization therapies after MI, such as percutaneous coronary intervention, a large proportion of patients will suffer progressive ventricular remodeling and eventually deteriorate to heart failure [2], which has driven numerous studies on heart regeneration after MI.

In recent years, accumulated studies are building a consensus that paracrine effects mediated by exosomes contribute to the major regenerative functions of cell therapies after myocardial injuries. Exosomes, the lipid bilayer nanovesicles ranging from 50 to 150 nm, intercellularly transfer signaling molecules such as proteins, mRNA, noncoding nucleic acids, lipids, and other cargoes to mediate cell–cell communication [3,4]. Exosomes released from various cell types, including mesenchymal stem cells (MSCs) [5,6], cardiac progenitor cells (CPCs) [7,8,9], and pluripotent stem cell (PSC)-derived cells [10,11] have been extensively studied for cardiac repair [12]. In addition to stem cell sources, exosomes originating from body fluids were also proved to have cardioprotective effects [13,14]. Exercise-induced serum or plasma exosomes were able to protect the heart by promoting CM survival via exosomal miRNAs, especially miR-342-5p [15,16]. Additionally, exosomes purified from rat or human plasma after remote ischemic pre-conditioning (RIC) treatment could provide an anti-apoptosis effect in CMs by the HSP70/TLR4 axis [17]. Therefore, plasma or serum exosomes, extracted from circulating blood under specific body states, might also be a potential therapeutic option for ischemic diseases. However, it is still confusing how the mixed bioactive molecules (RNAs, proteins, or others) in exosomes play their beneficial role on the recipient cell behaviors, which extremely hinders translational applications in the future [18].

Early neonatal mammals, including rodents and humans, retain a transient regeneration capacity after heart injury [19,20,21]. In addition to the endogenous proliferative ability of existing CMs, cumulative investigations have elucidated that non-cardiomyocytes (non-CMs), such as endothelial cells (ECs) [22,23] and macrophages [24,25], or the extracellular microenvironment [26,27] are complexly affecting cardiac regeneration in neonatal mammals [28]. Interestingly, the neonatal CPC exosomes showed significant repairing effects on heart function and structure after ischemic injury, while the CPC exosomes from older children failed [29]. Moreover, inspired by the discovery that plasma exosomes from young mice could enhance the physiological function and cognitive ability of old mice by delivering anti-aging substances [30], whether the plasma exosomes from neonatal mammals can contribute to non-CMs and transfer the regenerative benefits to the infarcted adult heart remains elusive. 

Hence, we delivered the plasma exosome of neonatal mice (npEXO) to the adult heart after acute myocardial infarction (AMI) and found that npEXO could structurally and functionally restore the infarcted heart to a certain extent. Cardiac EC was identified as the major non-CM cell type to receive npEXO ligand signals. A ligand–receptor network was constructed between npEXO and cardiac ECs, which might mediate the npEXO-induced angiogenesis and provide potential angiogenic targets for MI therapy.

## 2. Results

### 2.1. Plasma Exosomes from Neonatal Mice (npEXO) Facilitated the Structural and Functional Recovery of Adult Hearts after Acute Myocardial Infarction (AMI)

The npEXO was extracted from the plasma of neonatal mice (Postnatal day 7; P7). Nanoparticle tracking analysis (NTA) and transmission electron microscope (TEM) respectively showed that npEXO distributed from 50 to 150 nm in diameter and had a typical bilayer-membraned structure (Figure 1A,B and Appendix A). Two surface markers, CD63 and TSG101, were substantially expressed (Figure 1C).

To investigate the role of npEXO on cardiac repair, we treated the infarcted heart with npEXO by intramyocardial injection immediately after left anterior descending (LAD) artery ligation. Masson’s trichrome stain showed that 4 weeks after npEXO treatment, the size and shape of infarcted hearts were closer to those of the Sham group (Figure 1D), and the myocardial scar areas were significantly decreased (Figure 1E). We further labeled the CM cytomembranes with wheat germ agglutinin (WGA) and found that both in the border zone (BZ) and the remote zone (RZ), the cross-sectional sizes of CM were significantly smaller after npEXO treatment (Figure 1F–H). Furthermore, echocardiogram detection also showed that 4 weeks after AMI, npEXO administration ameliorated cardiac dilatation, indicated by the reduced LVIDd and LVIDs (Figure 1I,J and Appendix A), and improved cardiac systolic function, indicated by the elevated LVEF and LVFS (Figure 1K and Appendix A). In addition, the declined LV Vol-d and LV Vol-s also implied the mitigation of MI-induced heart failure (Appendix A). Moreover, we found that npEXO protected CMs from apoptosis (Appendix A), but had no significant impact on CM proliferation (Appendix A). Collectively, our results demonstrated that npEXO was able to structurally and functionally facilitate cardiac repair after AMI by promoting CM survival and alleviating cardiac remodeling.

### 2.2. Functional Enrichment and Pathway Analyses of the npEXO Proteins

Despite previous attention mostly having been focused on RNAs, exosomes are likely to exert function owing to their protein components [31]. Given that npEXO treatment contributed to adult heart repair after AMI, we further detected and analyzed the whole proteome to identify primary biological processes and pathways related to the cardioprotective effects provided by npEXO proteins. A total of 716 proteins were detected in npEXO by mass spectrometry. GO term analysis for biological process (BP) enrichment was mainly associated with wound healing, cell–substrate adhesion, extracellular matrix (ECM) organization, leukocyte migration, regulation of angiogenesis, activation of immune response, lipid transport, endothelial cell (EC) proliferation, EC differentiation, epithelial cell migration, etc. (Appendix A). Cellular component (CC) enrichment results included extracellular organelle, secretory granule, transport vesicle, and other components consistent with the features of circulating exosomes (Appendix A). Molecular function (MF) enrichment showed that the npEXO proteins were mainly involved in ECM structural constituent, integrin binding, cell adhesion molecule binding, phospholipid binding, etc., (Appendix A), which indicated that npEXO proteins might to some extent play their functions by interactions with recipient cells or extracellular environment components. Additionally, KEGG analysis suggested that the npEXO proteins were majorly enriched in signal pathways, such as ECM–receptor interaction, regulation of actin cytoskeleton, the PI3K-Akt signaling pathway, oxidative phosphorylation, and cell adhesion molecules (Appendix A). Taken together, the functional enrichment analysis of the npEXO proteome indicated that biological processes including angiogenesis, immune response, and cellular behaviors, such as EC proliferation, EC differentiation, and leukocyte and epithelial cell migration, might largely contribute to the regenerative effects of npEXO after AMI.

### 2.3. The Endothelial cells (ECs) as the Main Non-Cardiomyocyte (non-CM) Cell Type Was Responsible for Receiving npEXO Signals in the Mouse Heart

Exosomes are generally considered to promote heart repair by targeting non-CMs, which have been proven to play a crucial role in cardiac regeneration [12,32]. We then analyzed the published single-cell transcriptome data of the mouse hearts (neonatal (P4) and adult (8–12 weeks)) to further determine the major non-CM cell type to receive npEXO signals. After a similar quality control procedure according to their original study, the integrated data were normalized (Appendix A). Following the batch correction and principal component analysis, we performed an unsupervised clustering analysis and identified 16 distinct cell clusters using the t-SNE method (Figure 2A and Appendix A). Nine cell lineages were designated depending on their dominant expressed marker genes, including fibroblasts (FB, expressing Col3a1), endothelial cells (EC, expressing Cdh5), mural cells (MC, composed of smooth muscle cells and pericyte, expressing Myh11 and Kcnj8), macrophages (MAC, expressing Adgre1), monocytes (Mo, expressing Plac8), granulocytes (GN, expressing S100a8), B cells (expressing Ms4a1), T cells (T, expressing Cd3d), and natural killer cells (NK, expressing Nkg7) (Figure 2B and Appendix A). Although FB and EC were the dominant non-CM cell types, distinct non-CM heterogeneity existed in neonatal and adult hearts (Figure 2C). Then, the specifically expressed genes for each non-CM cell type were defined in neonatal and adult hearts, respectively (Figure 2D,E).

To investigate the communication network among npEXO ligand signals and non-CMs in the heart, we first developed an orthologs dataset containing 1316 ligands and 1516 receptors according to the cell–cell interactions database (Appendix A). After the intersection with our omics data, we identified 201 ligands in the npEXO, 246 non-CM receptors in the neonatal heart, and 302 receptors in the adult heart (Figure 2F–H). GO enrichment analysis showed that npEXO ligands and non-CM receptors (both neonatal and adult) collectively shared biological processes involved regulation of angiogenesis, EC and epithelial cell migration, cell adhesion, lymphocyte differentiation, and ECM organization (Figure 2I–K), implying that npEXO might actively affect the non-CM behaviors in both neonatal and adult hearts.

Based on the principle of paracrine effect mediated by ligand–receptor interaction to construct the communication network between npEXO and non-CMs, we first obtained 485 common processes between npEXO ligands and non-CM receptors in neonatal mouse heart (Appendix A). Specifically, 180 npEXO ligands and 197 non-CM receptors were involved. Among the non-CM receptor genes, ECs accounted for the highest proportion, followed by fibroblasts and macrophages (Appendix A), indicating that EC was the main non-CM cell type in neonatal mouse heart to receive npEXO ligand signals. Further, correlation analysis showed that these EC receptor genes were significantly associated with pro-regenerative processes or pathways during cardiac repair, such as regulation of angiogenesis and relevant cell behaviors, including EC or epithelial cell proliferation and migration, ERK1/ERK2 cascade, and cell–substrate adhesion (Appendix A).

Given that non-CM receptors in neonatal and adult mouse hearts nearly shared identical biological processes (Figure 2I,J), we speculated that EC was also the major non-CM cell type responsible for receiving npEXO ligands in the adult heart. We then analyzed the communication network between npEXO ligands and non-CM receptors in adult hearts and found 493 common biological processes were shared (Figure 3A), involving 181 npEXO ligands and 243 non-CM receptors. As expected, the npEXO ligands in the communication network were nearly the same genes and participated in biological processes alike to those of the neonatal heart, which were mainly involved in blood coagulation, ECM and collagen fibril organization, lipid transport, regulation of angiogenesis and EC migration, activation of immune response, cell–substrate adhesion, ERK1 and ERK2 cascade, and so on (Figure 3B). Similarly, the EC was still the top non-CM cell type to receive npEXO ligand signals, although the ratio of fibroblasts was increased in adult heart (Figure 3C,D). Correlation analysis of the EC receptors showed that several pro-regenerative processes mentioned above were found as well (Figure 3E). Therefore, our results indicated that cardiac EC was the main non-CM cell type in response to the npEXO ligand signals both in the neonatal and adult mouse heart, and the cardiac regenerative benefits of npEXO might largely owing to EC-mediated angiogenesis. 

### 2.4. npEXO Protected the Infarcted Hearts by Inducing EC Behaviors and Angiogenesis

We then tested whether the npEXO facilitated heart repair by promoting EC-mediated angiogenesis. First, to verify whether the npEXO act on cardiac ECs directly, we treated human umbilical vein endothelial cells (hUVECs) with npEXO and tested their pro-angiogenic behaviors separately by wound healing assay and tube formation assay. The results showed that after npEXO treatment, the hUVECs displayed faster migration in 10 h and 20 h (Figure 4A,B) and a better tube formation capacity (Figure 4C), indicated by the increased branch points number and tube length (Figure 4D,E). Furthermore, to measure the myocardial vascular density at the early stage (3 days) after AMI, we stained the vascular marker, CD31, and found that npEXO treatment could significantly promote angiogenesis both in the infarct zone (IZ) and BZ of the infarcted heart (Figure 5A,C,D). Similarly, staining of the arteriole marker, α-SMA, also confirmed the increased vascular density both in the areas of IZ and BZ after npEXO treatment (Figure 5B,E,F). In conclusion, our experimental results suggested that the npEXO promoted cardiac repair after AMI by their pro-angiogenic function and the direct impact on EC behaviors, including cell migration and tube formation.

### 2.5. Identification of the Common npEXO Ligand–Receptor Pairs in Both Neonatal and Adult Heart

According to the analysis of common biological processes shared by the npEXO ligands and cardiac non-CM receptors, we identified 181 npEXO ligands (in the adult) related to blood coagulation (Anxa5 and Klkb1), ECM and collagen fibril organization (Col14a1 and Comp), lipid transport (Lcat and Pltp), regulation of angiogenesis and EC migration (Ecm1 and Sparc), activation of immune response (C1qa and C8a), cell–substrate adhesion (Apod and Postn), ERK1 and ERK2 cascade (C1qtnf3, Fn1, and Slc9a3r1), regulation of endocytosis (B2m and Lbp) and regulation of peptidase activity (Fetub, Itih1, and Vcp) (Figure 3B). On the other hand, the cardiac EC receptor genes in the neonatal and adult heart represented for EC or epithelial cell migration (Cdh5, Kdr, and Nrp1), EC proliferation (Flt1, Tie1, and Dll4), cell–substrate adhesion (Cd36, Bcam, and Itga6), and regulation of angiogenesis (Aqp1, Ecscr, and Aplnr) were identified (Figure 3E and Appendix A).

We then proceeded to construct the communication relationship between npEXO and cardiac ECs based on the principle of ligand–receptor interaction. As the regenerative and angiogenic ability of the neonatal heart is superior to that of the adult heart [33], we first compared the total non-CM receptors between the neonatal and the adult heart. The 199 common receptor genes were mainly enriched in the GO terms of BP enrichment like granulocyte chemotaxis, leukocyte and epithelial cell migration, regulation of angiogenesis, proliferation of mononuclear cells, lymphocytes, and endothelial cells (Figure 6A,B), and processes including cell adhesion molecules, cytokine–cytokine receptor interaction, PI3K-Akt signaling pathway and ECM–receptor interaction by KEGG pathway analysis (Figure 6C). As expected, GO analysis suggested that specific receptors in the neonatal heart were largely associated with developmental processes, such as venous blood vessel development, stem cell differentiation, positive regulation of growth, mesenchyme development, and endothelial cell and leukocyte migration (Figure 6D), while the specific receptors in the adult heart were enriched by many immune-relevant processes, including cell chemotaxis, myeloid leukocyte activation, positive regulation of cytokine production, and lymphocyte-mediated immunity (Figure 6E).

Given that cardiac EC was the major non-CM cell type to receive npEXO ligand signals in both neonatal and adult hearts (Figure 3C,D, and Appendix A), we further constructed the communication networks among npEXO ligands and EC receptors, respectively, in the neonatal and adult heart according to the ligand–receptor dataset from the cell–cell interaction database and obtained a network of 114 interaction pairs formed by 63 npEXO ligands and 13 EC receptors in adult heart (Appendix A), and 118 interaction pairs formed by 55 npEXO ligands and 16 EC receptors in the neonatal heart (Appendix A). Among them, 66 interaction pairs formed by 41 npEXO ligands and 9 EC receptors appeared in both neonatal and adult hearts (Appendix A, Figure 6F,G), which might to some extent explain the role of npEXO in promoting angiogenesis during cardiac repair.

### 2.6. The Common Ligand–Receptor Pairs Shared by npEXO and MSC-EXO with Cardiac ECs

As one of the most widely studied exosomes in cardiovascular diseases, MSC-derived exosomes (MSC-EXO) were proven to be beneficial to cardiac regeneration and also played a pro-angiogenic role during heart repair [6,34,35]. Similarly, the MSC-EXO ligands were majorly received by ECs in the adult heart (Appendix A). Given these, we compared the npEXO ligands with those of MSC-EXO and found that 84 common proteins were shared (Figure 7A). GO term analysis for the BP enrichment showed that the common ligands shared by npEXO and MSC-EXO were mainly related to extracellular regulation of signal transduction, ECM organization, collagen metabolic process, cell–substrate adhesion, EC migration and proliferation, and wound healing and regeneration (Figure 7B). Besides, the CC enrichment included blood microparticle, collagen trimer, interstitial matrix, extracellular vesicle, etc. (Appendix A), and the MF enrichment was associated with ECM structural constituent, collagen binding, integrin binding, etc. (Appendix A). KEGG pathway analysis of the common proteins was associated with the ECM–receptor interaction, focal adhesion, the PI3K-Akt signaling pathway, and so on (Figure 7C). Particularly, in addition to typical angiogenesis-related processes (such as EC migration and proliferation), the common ligands were significantly enriched in ECM related components and processes, which were also considered to be critical for angiogenesis during cardiac regeneration [36,37,38,39], consistent with the enrichment results of ligands in npEXO and MSC-EXO, respectively (Figure 2H,K and Appendix A).

Next, we further identified 118 ligand–receptor pairs (formed by 61 ligands and 17 receptors) among the MSC-EXO ligands and the EC receptors of adult heart (Appendix A), in which there were 76 common pairs (formed by 36 ligands and 8 receptors) collectively shared by npEXO and MSC-EXO (Figure 7D,E and Appendix A). By intersecting these 76 common pairs with the 66 common pairs of npEXO shared by the neonatal and the adult heart, we obtained a final communication network of 48 interaction pairs formed by 28 ligands and 5 EC receptors, namely Itga6, Cd36, Scarb1, Kdr, and Tek (Figure 7F,G and Appendix A). Moreover, by re-analyzing the single-cell transcriptome data of seven healthy human hearts [40], we found that except SCARB1, the rest of four EC receptors (KDR, ITGA6, TEK, and CD36) are conservatively expressed in cardiac ECs (Appendix A). A large proportion of the 28 ligands were ECM proteins (Col18a1, Col4a2, Fbn1, Lamb1, Anxa5, etc.), which have been reported to play an instrumental role in angiogenesis by mediating ECs to sense and integrate the pro-angiogenic stimulus [36,38]. Other ligands included Clu (Clusterin), which has also been identified in exosomes originated from pericardial fluid and could facilitate arteriogenesis during heart regeneration after MI [14], and Hspg2, a kind of heparan sulfate proteoglycan, which served as an essential determinant of angiogenesis following hind-limb ischemia [41]. All in all, these communication networks constructed in our study to some extent unveil the potential molecules of npEXO in regulating EC behaviors and angiogenesis, and inspire the exploration of potential targets for pro-angiogenic therapy after MI (Appendix A).

## 3. Discussion

Exosomes originated from cells or other biological sources have proved to be a promising therapeutic option for cardiovascular diseases [12,13]. Recently, plasma or serum-derived exosomes under certain conditions, such as long-term exercise, could also provide cardioprotective and regenerative benefits during myocardial repair [15,42], but the underlying functional molecules of their cardioprotective role were still unclear. Previous omics analyses of exosomes mainly focused on RNAs, such as miRNA, lncRNA, and circRNA. The proteins in exosomes were recently considered to likely exert functional benefits, due to their biological concentration, biochemical functions, and proficiency to initiate timely biochemical reactions [31,43]. Here, we reported for the first time that by intramyocardial delivery into the adult heart immediately after AMI, exosomes from neonatal plasma could significantly reduce the scar area, ameliorate cardiac remodeling, and enhance ventricular systolic function, characterized by the improvement of multiple echocardiographic parameters, such as ejection fraction, which plays an important role in the follow-up of MI patients and serves as an independent predictor for in-hospital mortality of patients with cardiogenic shock [44,45]. In-depth proteome analysis suggested that the npEXO proteins might majorly promote angiogenesis by regulating the behaviors of recipient cells (such as ECs).

Most early research of cardiac regeneration focused on the direct regulation of CM proliferation and apoptosis, yet growing evidence found that non-CMs including ECs, cardiac fibroblasts, and immune cells also play a vital role in cardiac regeneration [46]. A recent study has unveiled the typical tempo-spatial distributions and highly synergistic interactions of non-CMs during the regenerative process in zebrafish hearts, and the disturbance of macrophage dynamics compromised cooperative interactions of non-CMs, thus impairing cardiac regeneration [32]. The two EC subtypes, capillary and lymphoid ECs, were also reported to respectively contribute to collateral artery formation and CM survival during adult heart repair [22,23]. Moreover, it was generally accepted that the primary heart regenerative mechanisms of exosomes were associated with stimulating angiogenesis, resisting CM apoptosis, alleviating inflammation, and reducing fibrosis, which were mostly mediated by non-CMs in the heart [12]. As we know, the neonatal mammal-like mice could successfully repair the damage heart in a short window, though whether the npEXO consistently functioned on non-CMs in neonatal and adult heart repair remained unexplored. Notably, we performed the single-cell transcriptomic analysis of non-CMs in both neonatal and adult mouse hearts, and recognized the ECs, rather than fibroblasts or immune cells, as the major non-CM cell type in both neonatal and adult hearts to receive npEXO ligand signals, implying that cardiac ECs do play a crucial role in the cardiac regenerative effects mediated by exosomes. Moreover, enrichment analysis showed that several critical pro-angiogenic processes including regulation of angiogenesis, EC migration, cell–substrate adhesion, ERK1/ERK2 cascade, etc., were collectively shared by npEXO ligands and non-CM receptors. Additionally, the pro-angiogenic effect of npEXO was further verified by vascular marker staining in vivo and EC migration and tube formation in vitro. Given that aging hearts show a decline in angiogenic capacity [47], it is noteworthy whether this npEXO would impact the pro-angiogenic effect in the optimal model for MI patients over the age of 50, which should be considered and investigated in the future. Together, our findings supported that npEXO directly acted on cardiac ECs and induced angiogenesis, which might serve as a pivotal reason to ameliorate the impaired heart after AMI.

Effective cardiac regeneration requires a timely and robust angiogenic response extending from the border zone to the core of the infarct zone. More importantly, restoration of a vigorous vascular network can provide the myocardium with necessary oxygen and nutrition, thereby reducing the secondary cell death of CMs and mitigating subsequent cardiac remodeling and deterioration in heart function [48]. Recently, a cytokine METRNL, originated from monocytes and macrophages, was identified as a high-affinity ligand for KIT receptor expressing on ECs, to drive mighty angiogenesis after MI [49]. The overexpression of VEGF-B, rather than VEGF-A, promoted EC proliferation and coronary vessel formation by binding the VEGFR-1 decoy receptor, thus significantly rescuing the infarcted adult heart in structure and function [50]. However, the complexity of angiogenesis and related cellular behaviors in the context of MI have not been completely understood. Since ligand–receptor interaction is the key manner to transduce extracellular signals into the cells and then affect their behaviors [51], we found that the npEXO ligands, which could be released into the extracellular space and directly recognized by cell-surface receptors of cardiac ECs, initiated the intracellular signaling cascade and prompted a timely pro-angiogenic response mediated by ECs. Our study further revealed a communication network of 48 interaction pairs containing 28 npEXO ligands, which might mainly mediate the pro-angiogenic effect of npEXO by recognizing five cardiac EC receptors (Kdr, Scarb1, Cd36, etc.). Endocardial overexpression of Kdr, also known as Vegfr2, could convert the adult endocardium into coronary vessels after MI, thus structurally and functionally improving the infarcted heart [52]. Additionally, the EC Kdr-mediated paracrine signaling contributed to physiological cardiac hypertrophy with angiogenesis [53]. Scarb1, namely scavenger receptor class B type I, could mediate high-density lipoproteins (HDLs) to rescue diabetes-impaired angiogenesis [54]. Another scavenger receptor, Cd36, which is remarkably abundant in cardiac ECs, could alleviate cardiac ischemia-reperfusion injury and has been considered to be a promising therapeutic target for a series of cardiovascular diseases [55]. These studies supported that the cardiac EC receptors mentioned above were crucial for angiogenesis in cardiovascular diseases, thus from the side confirming that their corresponding ligands, such as Clu and Hspg2, might serve as the key mediators for the pro-angiogenic effect of npEXO relying on a ligand–receptor interaction manner. In summary, the communication networks here provide an important clue to explore intervention targets for revascularization after MI.

Taken together, we for the first time reported that the plasma exosomes originating from neonatal mice could improve the infarcted adult heart, partly owing to their pro-angiogenic effect mediated by ECs, which might be explained by the interaction relationship of npEXO ligands and cardiac EC receptors (Figure 8). The ligand–receptor communication networks systematically established in our study may inspire the exploration of therapeutic targets in the future.

## 4. Materials and Methods

### 4.1. npEXO Isolation

Blood was collected from neonatal mice (postnatal day 7; P7) and centrifuged at 3000× *g* for 10 min at 4 °C to isolate plasma, then 10,000× *g* for 20 min at 4 °C to remove residual blood cells. A Hieff ^®^ Quick exosome isolation kit (for serum/plasma) was used for the isolation of plasma exosomes based on the manufacturer’s instructions (Yeasen Biotech, Shanghai, China). Each exosome sample was purified from 200 μL plasma (collected from 5 neonatal mice and EDTA-Na2 was used as an anticoagulant at the working concentration of 1.5 mg/mL).The isolated exosomes used for subsequent experiments and proteomic sequencing were further purified by 100 kD ultrafiltration tubes (Millipore, Boston, MA, USA) to centrifuge at 14,000× *g* for 20 min at 4 °C, then sterilized by 0.22 μM filters and diluted in sterile PBS to the final concentration of 400 μg/mL after being quantified by a BCA protein quantitative kit (TransGen Biotech, Beijing, China).

### 4.2. npEXO Characterization

The size distribution of exosomes was detected by nanoparticle tracking analysis (NTA) at the concentration ranging from 5.0 to 10.0 × 10^7^ particles/mL, including electrophoresis and Brownian motion video analysis operated by laser scattering microscopy ZetaVIEW (Particle Metrix, Meerbusch, Germany) and software (ZetaView 8.04.02 SP2); The exosome structure was observed by transmission electron microscope (TEM). Briefly, 20 μL exosome suspension was dropped into the copper grid and remained for 1 min, 2% phosphotungstic acid solution was used for negative dyeing and fixation for 5 min, then dried at RT and observed by TEM; two exosome surface markers, CD63 (1:2000; Yeasen Biotech, Shanghai, China) and TSG101 (1:1000; Yeasen Biotech, Shanghai, China) were detected by western blot.

### 4.3. Western Blot

Exosomes were lysed in RIPA buffer (Beyotime Biotech, Shanghai, China) with protease inhibitor cocktail (Roche). The protein concentration was quantified by Easy II Protein Quantitative Kit (BCA) (TransGen Biotech, Beijing, China) and 10 μg protein extracts were used to run on 10% SDS-polyacrylamide gel. After transferred, the PVDF membranes were incubated with primary antibodies against CD63, TSG101, and Albumin at 4 °C overnight, then incubated with corresponding secondary antibodies conjugated with horseradish peroxidase (1:2000; Thermo Fisher Scientific, Waltham, MA, USA) at RT for 1 h, and exposed by chemiluminescence.

### 4.4. Mouse Model of AMI and npEXO Treatment

Eight to ten-week-old C57BL/6N male mice were used for the construction of an adult AMI model as previously described [56]. In brief, mice were anaesthetized in a hermetic chamber with 3% isoflurane and performed thoracotomy. The close proximity of the left anterior descending (LAD) coronary artery was visualized and ligated by an 8-0 poly-propylene suture. Immediately after LAD ligation, 20 μg exosomes (sterilized by 0.22 μM filters and diluted in 50 μL PBS) or isochoric PBS were injected slowly by a 30-gauge insulin syringe, at 3 sites around the infarcted border. The thoracic cavity was closed by a 5-0 poly-propylene suture. Mice subjected to MI and intramyocardial injection were raised and processed for functional and structural detection at rational time points.

The general study design of animal experiments was as follows. For cardiac function assessment, transthoracic echocardiography was performed at 1, 2, and 4 weeks after AMI and npEXO treatment; three groups of mice were included; n = 3 for the Sham group, n = 3 for the AMI + PBS group, and n = 3 for the AMI + npEXO group, respectively. For cardiac structure assessment, mice were sacrificed at 4 weeks after AMI, and Masson’s staining and WGA staining were performed to respectively observe the cross-sectional scar areas and cell size of CMs; three groups of mice were included, n = 3 for the Sham group, n = 5 for the AMI + PBS group, and n = 5 for the AMI + npEXO group, respectively. For angiogenesis assessment, mice were sacrificed at 3 days after AMI, and the CD31 and α-SMA staining were performed to observe the vascular density; two groups of mice were included, n = 4 for the PBS group and n = 5 for the npEXO group, respectively.

### 4.5. Masson’s Staining

Four weeks after MI, the mice hearts were collected and cut into consecutive frozen tissue sections (0.3 mm interval), spanning from the occlusion point down to the apex. Myocardial and fibrotic tissues of the cardiac cross-section were simultaneously visualized by Masson’s staining according to the manufacturer’s instructions (Solarbio Science & Technology, Beijing, China), and statistics of the scar areas were performed by ImageJ 1.51e (Center for Information Technology, Bethesda, MD, USA) as previously described [57].

### 4.6. WGA Staining and Immunofluorescence Staining

Four weeks after MI, wheat germ agglutinin (WGA; 1:100; Thermo Fisher Scientific, Waltham, MA, USA) was used to label the CM membrane and statistics of the cross-sectional CM cell size in the BZ and RZ were performed by ImageJ 1.51e. 3 days after MI, blood vessels in the IZ and BZ were stained by vascular marker, CD31 (1:500; ABclonal Technology, Wuhan, China); arteriole marker, α-SMA (1:500; Cell Signaling Technology, Boston, MA, USA) and corresponding fluorescent secondary antibodies (1:2000; Thermo Fisher Scientific, Waltham, MA, USA). Vascular density was calculated by ImageJ 1.51e with a plugin (VesselJ) from ImageJ official toolsets.

### 4.7. Transthoracic Echocardiography Detection

Cardiac functional assessments were conducted by a Vevo2100 micro-ultrasound system and MS-550S probe (VisualSonics, Toronto, ONT, Canada), respectively at 3 time points after MI and npEXO treatment (1, 2, and 4 weeks). Two-dimensional M-mode images were captured from a parasternal short-axis view of the left ventricle at the level of papillary muscles, and left ventricular internal diameters at end diastole (LVIDd) and end systole (LVIDs) were measured based on the M-mode recordings. Left ventricular ejection fraction (LVEF), fractional shortening (LVFS), left ventricular volume at end diastole (LV Vol-d), and end systole (LV Vol-s) were calculated from at least three separate contraction cycles of each mouse by equations reported previously [58].

### 4.8. hUVEC Migration and Tube Formation Assay

The human umbilical vein endothelial cells (hUVECs) monolayers were cultured in 24-well plates (Thermo Fisher Scientific, Waltham, MA, USA) and scratched with a 200 μL pipet tip, then washed by PBS and incubated with culture medium (RPMI1640 with 10% FBS), containing 400 μg/mL sterilized exosomes or isochoric PBS. The treated cells were respectively snapped at 0, 10, and 20 h. The percentage of wound closure represented the ratio of area covered by cells vs. total area, measured by ImageJ. For tube formation assay, 8.0 × 10^4^/well hUVECs were seeded on a 24-well plate coated with matrigel (BD Biosciences, Franklin Lakes, NJ, USA), and observed 6 h later. The number of branch points and tube length were calculated by ImageJ 1.51e with the Angiogenesis Analyzer plugin from ImageJ official toolsets.

### 4.9. Mass Spectrometry Analysis of npEXO Proteins

Blood samples stored at −80 °C were thawed at 25 °C water bath, and then exosome isolation was performed as mentioned above. After extraction and quantification by a BCA kit, the proteins were digested overnight with 50:1 trypsin (trypsin-to-protein mass ratio). Then the digested peptides were desalted by a C18 column. The qualitative and quantitative analysis of exosomal proteins were performed on an Orbitrap Fusion Lumos mass spectrometer (Thermo Scientific, Waltham, MA, USA) coupled with EASY-nLC 1000 liquid chromatography system (Thermo Scientific, Waltham, MA, USA). The vacuum-dried samples were reconstituted with 0.1% formic acid (FA), and separated on a 75 μm I.D. × 25 cm C18 analytical column (C18, 1.9 μm, 120 Å, Dr. Maisch GmbH, Tubingen, Germany) with a mobile solution flow rate of 200 nL/min. The gradient elution program was as the following: 2-8% solvent B (100% acetonitrile and 0.1% formic acid) in 5 min, 8-24% solvent B in 85 min, 24-32% solvent B in 20 min, 32-90% solvent B in 5 min, keep 90% solvent B for 5 min. Data were acquired with full scans (m/z 350-1800) at a mass resolution of 60,000 (FWHM). For MS2, the normalized automatic gain control (AGC) target was 1 × 10^6^ with a resolution of 15,000 and a maximum injection time of 22 ms, and the precursor ions were fragmented in the high-energy-collision dissociation (HCD) cell at normalized collision energy (NCE) of 30%.

### 4.10. Pre-Analysis of Proteome Data

The raw files obtained from Orbitrap Fusion Lumos mass spectrometer were analyzed by Proteome Discoverer 2.4 (Thermo Scientific, Waltham, MA, USA), and the database was Swissprot (mouse). Search conditions: Fixed modification was set as Carbamidomethyl, and the variable modification was set as: Methionine oxidation (M), protein N-terminal acetylation (Acetyl). Trypsin digestion was selected to allow two missing sites. The maximum mass errors of mother and daughter ions were set at 20 ppm and 0.5 Da, respectively. The false positive rate (FDR) of peptide segment was less than 5%. The PXD020948 dataset, containing exosome proteins derived from human bone marrow (MSC-EXO), was obtained from the publicly proteomic data (ProteomeXchange) [31]. The common proteins of MSC-EXO and npEXO and the specific proteins were obtained for the following analysis.

### 4.11. scRNA-Seq Data Analysis

The two datasets, GSE153481 and e-matb-7376, were downloaded from the GEO database (accessed on 5 October 2022, https://www.ncbi.nlm.nih.gov/geo/) or the ArrayExpress database (accessed on 5 October 2022, https://www.ebi.ac.uk/arrayexpress/), which contained the scRNA-seq data of non-CMs in neonatal and adult (8–12 weeks) mouse hearts, respectively. These two datasets were further analyzed as follows.

Cell filtering: Seurat R package [59] was used for analyzing the scRNA-seq data of non-CMs in neonatal and adult mouse hearts. We tested several criteria to filter cells from two gene expression matrices. Common quality control filtering metrics were applied as follows in both datasets. The cells that expressed less than 200 or more than 4000 unique genes were filtered out. Genes that were expressed in less than 3 cells were filtered out. Specifically, neonatal mouse heart cells with over 25% of raw UMIs mapping to mitochondrial genes were filtered out, while 15% was set as the limitation for adult mouse heart cells. In total, 9155 cells were kept for downstream analysis after merging filtered datasets.

Single-cell clustering: The FindVariableGenes function of Seurat was used to select 2500 high-variable genes and the canonical correlation analysis (CCA) method was applied to minimize the batch effect. Then, by conducting principal component analysis (PCA), 20 principal components (PCs) were used for t-distributed stochastic neighbor embedding (t-SNE) dimensionality reduction. The FindClusters function based on the shared nearest neighbor (SNN-Cliq) clustering method was used with default parameters (k = 30). SNN clustering with a resolution value of 0.5 identified 16 cell clusters.

Cell type annotation: The FindAllMarkers function of Seurat was used to identify featured genes for each cluster with parameters test.use = wilcox, min.pct = 0.25, thresh.use = 0.25, only.positive = TRUE and return.thresh = 0.05. Each cluster was annotated to specific cell types by the expression of globally known marker genes in the heart, including fibroblasts, endothelial cells, smooth muscle cells, pericytes, macrophages, monocytes, granulocytes, B cells, T cells, and natural killer cells.

### 4.12. Preparation of Ligand and Receptor Datasets

The cell–cell interactions filtered the existing curated human protein–protein interaction (PPI) data, involving tables of ligands, receptors, and ligand–receptor interaction sets. The homologene package was used to develop mouse orthologs datasets of ligands and receptors. Then intersect function was used for the following identification of ligands and receptors by the corresponding dataset. The exosome ligands were generated from the intersection of ligands dataset and exosome proteins including npEXO and MSC-EXO. In order to define specifically expressed genes for each cell type, we used the FindAllMarkers function of Seurat with parameters test.use = wilcox, min.pct = 0.25, thresh.use = 0.25, only.positive = TRUE and return.thresh = 0.05 in neonatal and adult mouse hearts, respectively. The possible receptors of neonatal and adult mouse hearts were selected by intersecting receptors dataset and cell-type-specific genes that were filtered with an adjusted *p*-value of 0.01. The ligand–receptor pairs were used for identifying the corresponding relationships between exosome ligands and non-CM receptors in the heart.

### 4.13. Functional Enrichment Analysis

The clusterProfiler package of R was utilized for GO enrichment analysis, containing biological process (BP), cellular component (CC), molecular function (MF), and KEGG enrichment analysis. Adjust *p*-value <0.05 was regarded as the cut-off criteria in functional enrichment analysis. Multiple vectors of genes were imported for functional enrichment analysis, including npEXO proteins, npEXO ligands, and the cell-type-specific receptors in neonatal and adult mouse hearts. In addition to direct analysis of ligands and receptors, celltype-specific receptors of neonatal mouse heart non-CMs were compared with those of adult mouse heart non-CMs. Similarly, npEXO ligands were compared with MSC-EXO ligands. Enrichment analysis was also performed in the common or specific receptors of non-CMs from neonatal and adult mouse hearts, and common ligands of exosomes from MSC-EXO and npEXO. The representative results of functional enrichment analysis were visualized using the ggplot2 and enrichplot package of R.

### 4.14. Construction of Exosome-Heart Interaction Network

To build a communication network that exosome ligands directly influence the non-CM function by receptors in the heart, we used a cross-match method [60]. Crucially, we compared BP enrichment results of exosome ligands and non-CM receptors in the heart. The overlapping terms might refer to the global interaction network of exosome ligands and non-CM receptors in the heart. Then we extracted ligands and receptors participating in the global interaction network. Three communication networks were built, including npEXO with non-CMs of the neonatal or adult mouse heart, and MSC-EXO with non-CMs of the adult mouse heart. The representative BPs and ligands were drawn using the R package pheatmap. As for the receptors, the DoHeatmap function of Seurat was utilized for visualizing the expression, and the contribution level of receptor in each cell type was drawn using the ggplot package. The communication network was visualized by Cytoscape. The UpSetR package of R was used for the comparison of different communication networks.

### 4.15. Statistical Analysis

Data were presented as mean ±SEM. Differences between the two independent groups were evaluated by Student’s t-test. Differences among more than two groups were evaluated by analysis of variance (ANOVA) tests. All statistical analyses were conducted by GraphPad Prism 7.0 (GraphPad Software, San Diego, CA, USA). Statistical differences were considered significant when *p* < 0.05 (* represented for *p* < 0.05; ** represented for *p* < 0.005; *** represented for *p* < 0.0005; **** represented for *p* < 0.0001).

## 5. Conclusions

Altogether, we found that npEXO could to some extent restore the infarcted adult heart in structural and function. And the npEXO ligand signals were mainly received by cardiac ECs to induce angiogenesis. The ligand–receptor network constructed in our study might provide potential targets for cardiac repair following MI.

## Figures and Tables

**Figure 1 ijms-24-03196-f001:**
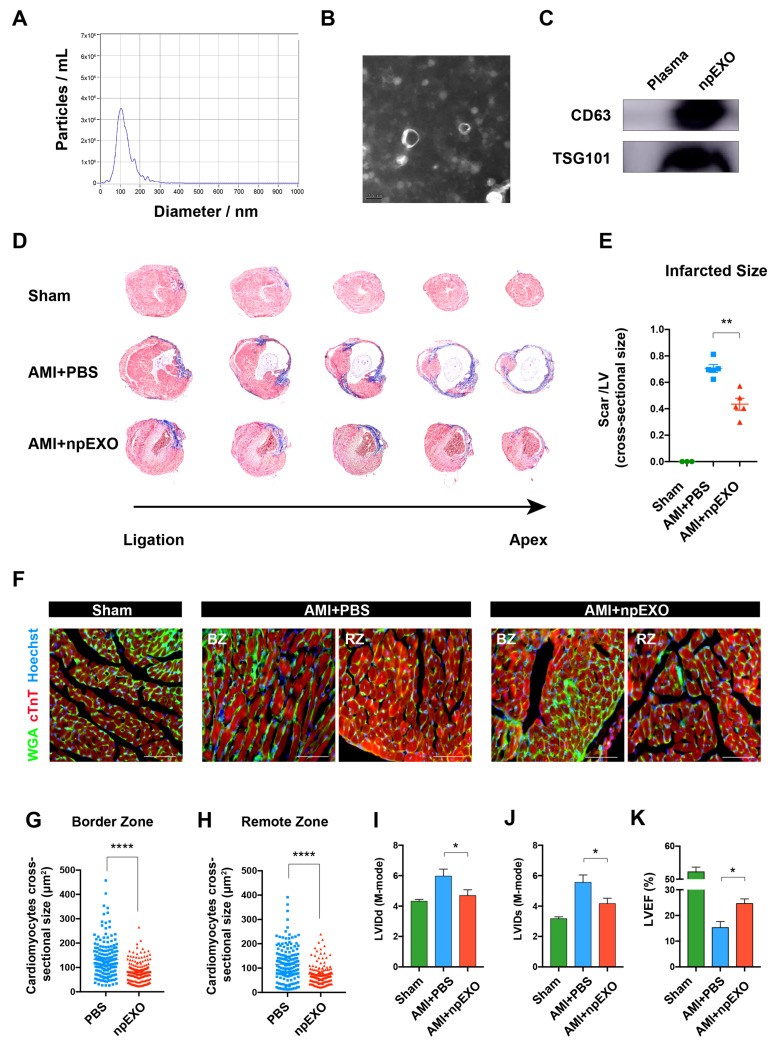
Plasma exosomes from neonatal mice (npEXO) structurally and functionally restored the adult heart after acute myocardial infarction (AMI). (**A**) npEXO size distribution was detected by NTA. (**B**) npEXO structure and size detected by TEM (Scale bar, 100 nm). (**C**) npEXO surface markers (CD63 and TSG101) detected by western blot. (**D**) Masson’s staining of mouse hearts 4 weeks after AMI and npEXO treatment. (**E**) Statistics of infarcted size (Sham, n = 3; AMI + PBS, n = 5; AMI + npEXO, n = 5). (**F**) CM membranes labeled by WGA staining 4 weeks after AMI and npEXO treatment (Scale bars, 100 μm). (**G**,**H**) Statistics of CM cross-sectional size in the border zone (BZ) (**G**) and remote zone (RZ) (**H**). (**I**–**K**) Statistics of cardiac function indicators LVIDd (**I**), LVIDs (**J**), and LVEF (**K**) at 4 weeks after AMI (Sham, n = 3; AMI + PBS, n = 3; AMI + npEXO, n = 3). (* represented for *p* < 0.05; ** represented for *p* < 0.0005; **** represented for *p* < 0.0001).

**Figure 2 ijms-24-03196-f002:**
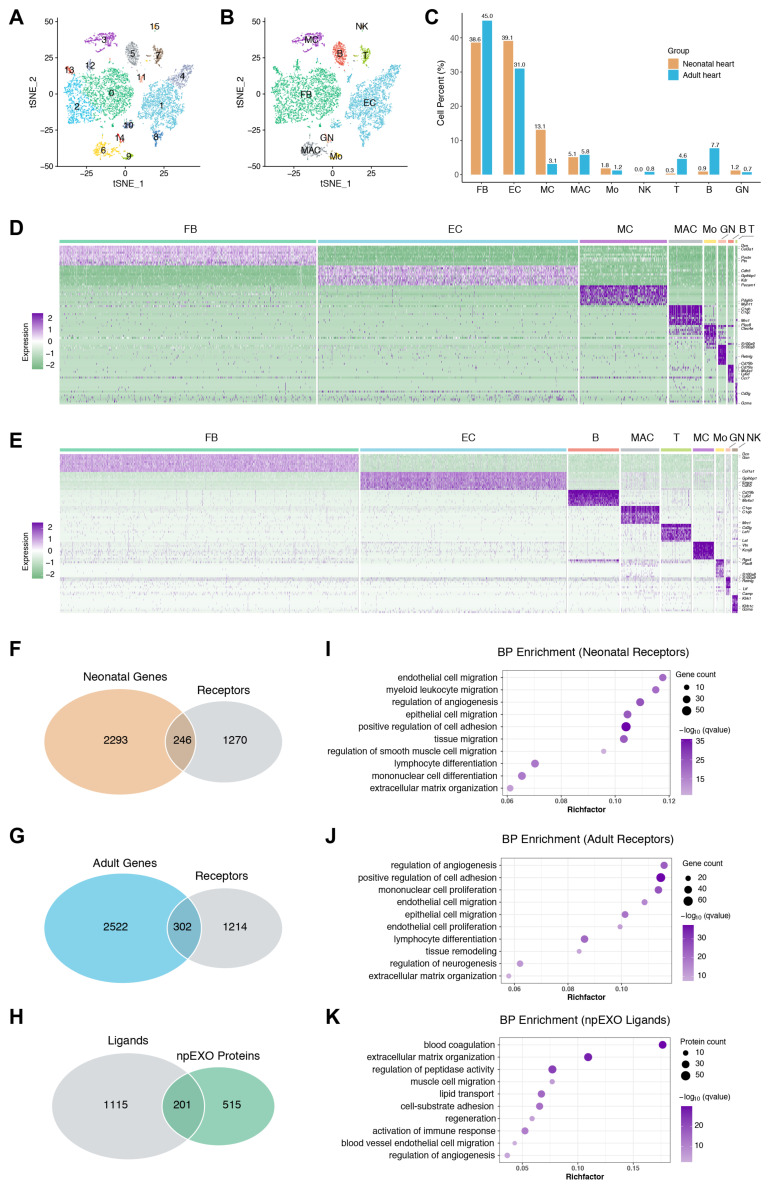
Single-cell transcriptome analysis reveals the non-CM receptors correlated with npEXO ligands in the neonatal and adult heart. (**A**) tSNE representation and separation of 9155 cells of the heart into sixteen clusters. Cells were marked by cluster number. (**B**) tSNE plot of non-cardiomyocytes (non-CMs) pooled from the neonatal and adult mouse hearts, showing the non-CM populations identified based on canonical marker genes. (**C**) Fraction of each non-CM population relative to all cardiac non-CMs in the neonatal and adult heart, respectively. (**D**,**E**) Heatmap of the top ten differentially expressed genes in each non-CM population of neonatal (**D**) and adult (**E**) heart. (**F**,**G**) Venn diagram of the non-CM genes of neonatal (**F**) and adult (**G**) hearts against the receptor dataset. (**H**) Venn diagram of npEXO proteins against the ligand dataset. (**I**,**J**) GO analysis (BP enrichment) of non-CM receptor genes in the neonatal (**I**) and adult (**J**) heart. (**K**) GO analysis (BP enrichment) of npEXO ligand proteins.

**Figure 3 ijms-24-03196-f003:**
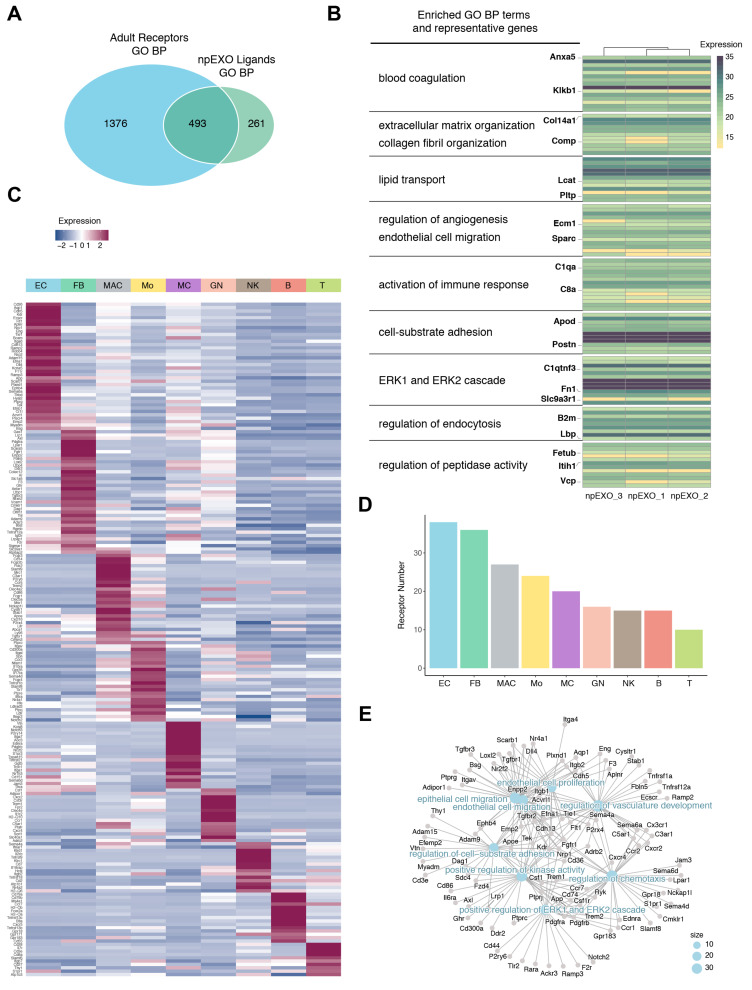
Construction of communication network between npEXO ligands and non-CMs in the adult heart. (**A**) Venn diagram showing GO results (BP enrichment) of npEXO ligands against those of non-CM receptors in the adult heart. (**B**) Heatmap of the npEXO ligands involved in npEXO and adult non-CM communication network. Clusters are assembled by GO analysis. The significantly enriched items and representative genes are shown on the left. (**C**) Heatmap displaying the expression of receptor genes involved in npEXO and non-CM communication network of the adult heart. (**D**) Sum of all communication network receptor genes expressed in each non-CM population. (**E**) Correlation analysis showed receptors expressed in endothelial cells (ECs) and pathways participated in a communication network.

**Figure 4 ijms-24-03196-f004:**
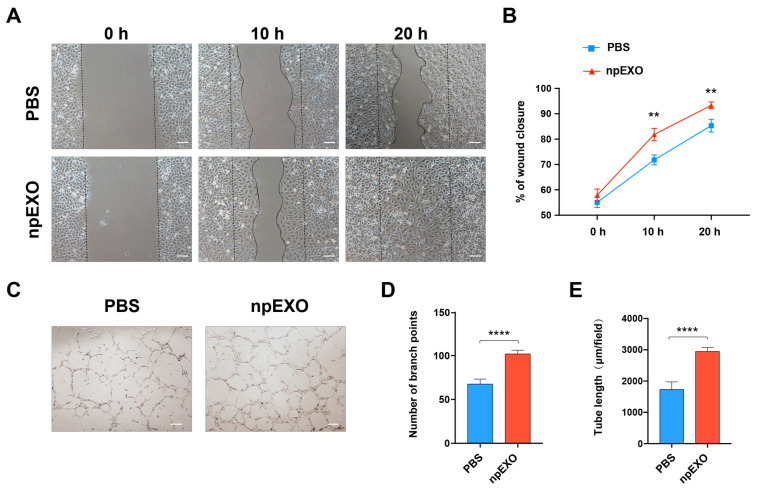
npEXO promoted EC migration and tube formation in vitro. (**A**) EC migration after npEXO treatment detected by wound healing assay. (Scale bars, 100 μm) (**B**) Statistics of wound closure fraction. (**C**) EC tube formation assay after npEXO treatment. (Scale bars, 200 μm) (**D**,**E**) Statistics of branch point number (**D**) and tube length (**E**). Three independent repeated experiments were conducted. (** represented for *p* < 0.005; **** represented for *p* < 0.0001).

**Figure 5 ijms-24-03196-f005:**
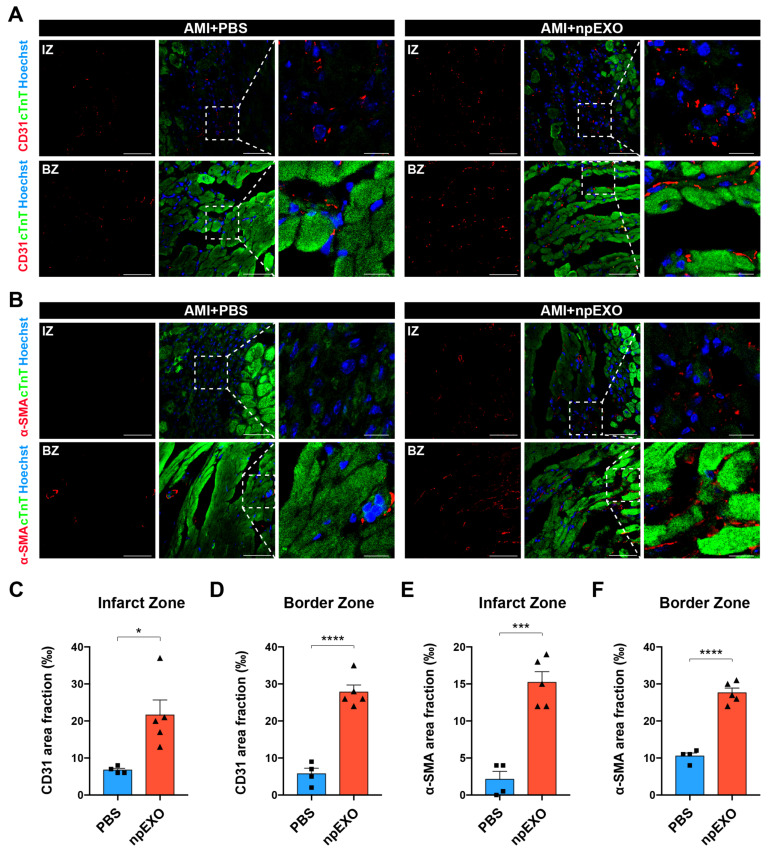
npEXO protected the infarcted hearts by inducing angiogenesis. (**A**) Staining of EC marker (CD31) 3 days after AMI and npEXO treatment. (**B**) Staining of SMC marker (α-SMA) 3 days after AMI and npEXO treatment (Scale bars, 100 μm). Scale bars for (**A**) and (**B**), 100 μm (left and middle panels of each group), 25 μm (right panels of each group). (**C**,**D**) Statistics of CD31 area fraction in the infarct zone (IZ) (**C**) and BZ (**D**) (PBS, n = 4; npEXO, n = 5). (**E**,**F**) Statistics of α-SMA area fraction in the IZ (**E**) and BZ (**F**) (PBS, n = 4; npEXO, n = 5). (* represented for *p* < 0.05; *** represented for *p* < 0.0005; **** represented for *p* < 0.0001).

**Figure 6 ijms-24-03196-f006:**
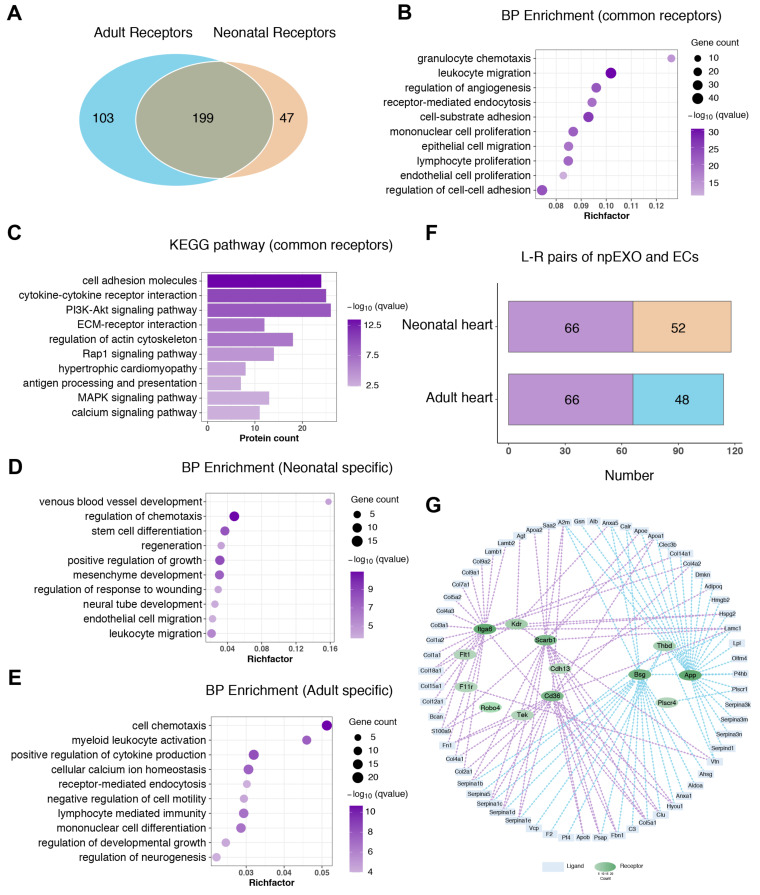
Identification of the common npEXO ligand–receptor pairs shared by the neonatal and adult heart. (**A**) Venn diagram of total non-CM receptors in the neonatal heart against those in the adult heart. (**B**,**C**) GO analysis (**B**) and KEGG pathway (BP enrichment) (**C**) of common non-CM receptors shared by the neonatal and adult heart. (**D**,**E**) GO analysis (BP enrichment) of specific non-CM receptors respectively in the neonatal heart (**D**) and the adult heart (**E**). (**F**) The L-R pairs number of npEXO and ECs in the neonatal and adult heart (L-R pairs: ligand–receptor pairs; purple represented for the common part). (**G**) The communication network constructed among npEXO ligands and EC receptors in the neonatal and adult heart (purple lines represented for the common part and blue lines for the part specific in the adult heart; the hierarchical color depth of the receptors was related to the number of interactions).

**Figure 7 ijms-24-03196-f007:**
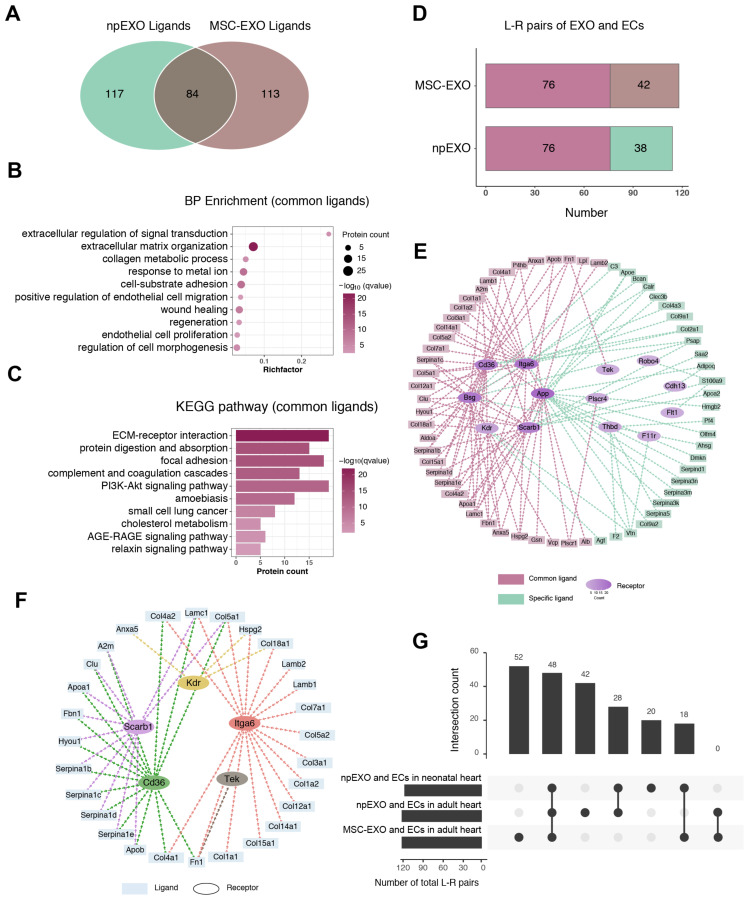
Identification of common ligand–receptor pairs shared by npEXO and MSC-EXO. (**A**) Venn diagram of total npEXO ligands against those of MSC-EXO. (**B**) GO analysis (BP enrichment) of common ligands shared by npEXO and MSC-EXO. (**C**) KEGG pathway analysis of common ligands shared by npEXO and MSC-EXO. (**D**) The L-R pairs number of npEXO (or MSC-EXO) and ECs in the adult heart (L-R pairs: ligand–receptor pairs; dark red represented the common part). (**E**) The communication network constructed among npEXO (or MSC-EXO) ligands and EC receptors in the adult heart (dark red represented for the common part and green for the npEXO-specific part; the hierarchical color depth of the receptors was related to the number of interactions) (**F**) The communication network represented for the ligand–receptor interactions overlapped between the common part shared in the neonatal and adult heart, and another common part shared by npEXO and MSC-EXO. (**G**) Venn diagram of the ligand–receptor pairs among three groups (npEXO and ECs in neonatal heart; npEXO and ECs in adult heart; MSC-EXO and ECs in the adult heart).

**Figure 8 ijms-24-03196-f008:**
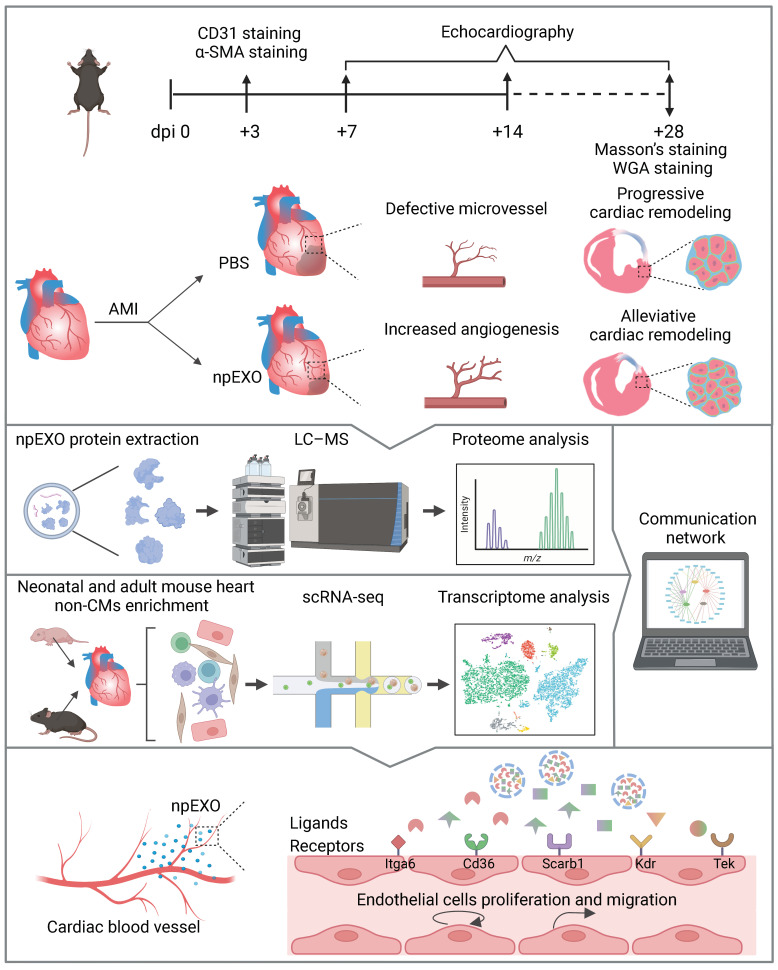
The schematic flow of experiments and bioinformatics analysis in this study. Three days after AMI and npEXO administration, CD31 and α-SMA staining were performed for angiogenesis assessment. Echocardiography was detected for cardiac function assessment at 7, 14, and 28 days post-MI. For cardiac structure assessment, Masson’s and WGA staining were conducted at 28 days post-MI (dpi means days post-MI). The npEXO administration increased angiogenesis in the infarcted adult heart (both IZ and BZ) and alleviated the subsequent cardiac remodeling (reduced scar area, smaller cardiac chambers, and CM cell size) (**Upper**). The workflow of constructing a communication network among npEXO ligands and non-CM receptors by combining proteome and single-cell transcriptome data (**Middle**). A final communication network of 48 interaction relationships was formed by 28 npEXO ligands and 5 cardiac EC receptors (**Lower**). (Created with Biorender.com).

## Data Availability

The npEXO proteome data have been deposited to the ProteomeXchange Consortium via the iProx repository with the data set identifier PXD038805. The MSC-EXO proteome data (PXD020948) was taken from the published study by Wang et al. [31]. The non-CM scRNA-seq datasets of neonatal and adult mouse heart were taken from GSE153481 [61] and e-mtab-7376 [62], respectively.

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
