# Peer review of "Neonatal Plasma Exosomes Contribute to Endothelial Cell-Mediated Angiogenesis and Cardiac Repair after Acute Myocardial Infarction"

_ijms, 2023, doi:10.3390/ijms24043196_

Round 1

Reviewer 1 Report

The manuscript presents a very nice a complete analysis of the potential of neonatal plasma exosome to treat Acute myocardial infarction. Several studies in vitro and in vivo analysis together with proteomics and scRNA-seq datasets are combined to identify pathways to explain the potential pro-angiogenic effects. However, some information is missing as well as limitations of the current study:

There is not manuscript section explaining the study design for the AMI model, the sample size is only indicated in Figure 1 (Sham=3, AMI+PBS=5, AMI+npEXo=5 for Masson´s stain; but for cardiac function at 4 weeks Sham=3, AMI+PBS=3, AMI+npEXo=3). The design should be included at least in the method section 4.4. Mouse model of AMI and npEXO treatment.  And even to include in Figure 8 (flow of experiments)

The mice model used seems younger (8-10 weeks-young mature) that the optimal for human AMI model (>50 years old), could this impact pro-anxiogenics?, I consider it should be discussed or present as limitation.

It is also remarkable that the npEXO comes from only single mice, so I wonder if individual variability could have an impact on potentiality. I consider it should be discussed

A better explanation of the heart function should be added in the paragraph of results: 2.1. npEXO facilitated the structural and functional recovery of adult hearts after AMI. The LV parameters are only indeed in Figure 1, I consider it is worth to better explain the graphics and results.

Author Response

Response to Reviewer 1 Comments

The manuscript presents a very nice a complete analysis of the potential of neonatal plasma exosome to treat Acute myocardial infarction. Several studies in vitro and in vivo analysis together with proteomics and scRNA-seq datasets are combined to identify pathways to explain the potential pro-angiogenic effects. However, some information is missing as well as limitations of the current study:

Point 1. There is not manuscript section explaining the study design for the AMI model, the sample size is only indicated in Figure 1 (Sham=3, AMI+PBS=5, AMI+npEXo=5 for Masson´s stain; but for cardiac function at 4 weeks Sham=3, AMI+PBS=3, AMI+npEXo=3). The design should be included at least in the method section 4.4. Mouse model of AMI and npEXO treatment. And even to include in Figure 8 (flow of experiments).

Response 1: We sincerely thank the reviewer to point out this issue. We have added description of the study design in the method section 4.4. Mouse model of AMI and npEXO treatment. In addition, a flow chart of animal experiments has been included in the upper part of Figure 8.

Point 2. The mice model used seems younger (8-10 weeks-young mature) that the optimal for human AMI model (>50 years old), could this impact pro-anxiogenics? I consider it should be discussed or present as limitation.

Response 2: We appreciate the reviewer’s suggestion. According to previous literatures, young adult mice (8-10 weeks) are usually used for constructing the MI model and investigating the mechanisms of cardiac regeneration, because they are more tolerant to the injury and easier to survive following MI than the old mice. The mouse model indeed cannot completely mimic human AMI patients over 50 years old or the spontaneous MI during aging. Given the decline of angiogenic capacity in the aging hearts[1], whether this would impact the pro-angiogenic effect induced by exosomes should be really considered and deserved to be further investigated in the future. We have included and discussed this in the revised manuscript. In addition, by re-analyzing the single-cell transcriptome data of 7 human hearts [2], we have found that except Scarb1, four of the EC receptor genes (Kdr, Itga6, Tek, and Cd36) proposed in our study are conservatively expressed in cardiac ECs of healthy people over the age of 50. Theoretically, we speculated that the exosomal ligands might also mediate their pro-angiogenic effect by recognizing these four cardiac EC receptors in the aging human hearts. We have added a supplemental figure (Supplemental Figure S8) and related description in result 2.6. The common ligand-receptor pairs shared by npEXO and MSC-EXO with cardiac ECs.

Point 3. It is also remarkable that the npEXO comes from only single mice, so I wonder if individual variability could have an impact on potentiality. I consider it should be discussed.

Response 3: Thanks for this thoughtful consideration. We are sorry to confuse you with inadequate details about the npEXO preparation. Actually, each npEXO sample for subsequent experiments was extracted from 5 neonatal mice, given that only about 50 μL plasma could be collected from a single neonatal mouse (P7). We have provided more detailed information in the method section 4.1. npEXO isolation.

Point 4. A better explanation of the heart function should be added in the paragraph of results: 2.1. npEXO facilitated the structural and functional recovery of adult hearts after AMI. The LV parameters are only indeed in Figure 1, I consider it is worth to better explain the graphics and results.

Response 4: We genuinely thank the reviewer’s advice. We have added some descriptions to better explain the echocardiographic parameters and their implications in the result section 2.1. npEXO facilitated the structural and functional recovery of adult hearts after AMI.

References

1. Edelberg, J. M. et al. Platelet-derived growth factor-AB limits the extent of myocardial infarction in a rat model: feasibility of restoring impaired angiogenic capacity in the aging heart. Circulation 105, 608-613, doi:10.1161/hc0502.103672 (2002).

2. Litviňuková, M. et al. Cells of the adult human heart. Nature 588, 466-472, doi:10.1038/s41586-020-2797-4 (2020).

Reviewer 2 Report

I have reviewed the manuscript entitled 'Neonatal Plasma Exosomes Contribute to Endothelial Cell-Mediated Angiogenesis and Cardiac Repair After Acute Myocardial Infarction'.

The introduction part should be shortened in order to provide fluency

The potential role of these exosomes should be discussed in the final section. The role of ejection fraction is very important in the follow up of patients with myocardial infarction. Therefore these exosomes following phase 3 trials can potentially be used in patients with myocardial infarction. Consider citing' The predictive value of age, creatinine, ejection fraction score for in-hospital mortality in patients with cardiogenic shock' and 'Predictors of In-Hospital Mortality in Patients With ST-Segment Elevation Myocardial Infarction Complicated With Cardiogenic Shock'

Author Response

Response to Reviewer 2 Comments

I have reviewed the manuscript entitled 'Neonatal Plasma Exosomes Contribute to Endothelial Cell-Mediated Angiogenesis and Cardiac Repair After Acute Myocardial Infarction'.

Point 1. The introduction part should be shortened in order to provide fluency.

Response 1: We thank for the reviewer’s suggestion. And we have modified the introduction section in our revised manuscript to make it concise and fluent.

Point 2. The potential role of these exosomes should be discussed in the final section. The role of ejection fraction is very important in the follow up of patients with myocardial infarction. Therefore these exosomes following phase 3 trials can potentially be used in patients with myocardial infarction. Consider citing' The predictive value of age, creatinine, ejection fraction score for in-hospital mortality in patients with cardiogenic shock' and 'Predictors of In-Hospital Mortality in Patients With ST-Segment Elevation Myocardial Infarction Complicated With Cardiogenic Shock'.

Response 2: We are grateful to the reviewer for this valuable advice. We have learned the proposed articles [1, 2], which suggested that ejection fraction was a very important predictor for in-hospital mortality of patients complicated with cardiogenic shock and played a significant role in the follow-up of MI patients. And we have discussed the potential role of npEXO in ejection fraction improvement in the discussion part of the revised manuscript.

References

1. Çinar, T. et al. The predictive value of age, creatinine, ejection fraction score for in-hospital mortality in patients with cardiogenic shock. Coronary artery disease 30, 569-574, doi:10.1097/mca.0000000000000776 (2019).

2. Hayıroğlu, M. et al. Predictors of In-Hospital Mortality in Patients With ST-Segment Elevation Myocardial Infarction Complicated With Cardiogenic Shock. Heart, lung & circulation 28, 237-244, doi:10.1016/j.hlc.2017.10.023 (2019).

Reviewer 3 Report

The authors describes that they intramyocardially delivered the plasma exosome of neonatal mice to the adult heart after MI and found that npEXO could structurally and functionally restore the infarcted heart. Also, cardiac endothelial cell was identified as the major non-cardiomyocyte cell type to receive npEXO ligand signals. I think the topic is relevant and it addresses the gap in this field by answering above main question.

The effects and mechanism the roles of exosomes in cardiac regenerative therapy is still unclear. This manuscript revealed that cardiac endothelial cell is identified as the major non-cardiomyocyte cell type to receive the exosome ligand signals which partly mediates the angiogenesis during adult heart repair.  

Regarding methodology, I think the number of mouse which was used for animal surgery is too small (n=3-5). Regarding controls, if they make different amount of npEXO injection or different number of spots where they inject the npEXO, it might be better.

I think conclusions are appropriate and it addresses the main question.

In many of the references, they wrote only one author and et al.

If they could add scale bars in Figure 4 A, C, that might be helpful.

Author Response

Response to Reviewer 3 Comments

The authors describes that they intramyocardially delivered the plasma exosome of neonatal mice to the adult heart after MI and found that npEXO could structurally and functionally restore the infarcted heart. Also, cardiac endothelial cell was identified as the major non-cardiomyocyte cell type to receive npEXO ligand signals. I think this manuscript is interesting and worthwhile to be published in this journal.

Response: We sincerely appreciate the reviewer’s positive comment on our study. And we have checked and revised our manuscript for further improvement.